# Parvaxanthines D–F and Asponguanosines C and D, Racemic Natural Hybrids from the Insect *Cyclopelta parva*

**DOI:** 10.3390/molecules26123531

**Published:** 2021-06-09

**Authors:** Heng Chen, Yong-Ming Yan, Dai-Wei Wang, Yong-Xian Cheng

**Affiliations:** 1Guangdong Pharmaceutical University, Guangzhou 510006, China; hengchen2021@foxmail.com; 2School of Pharmaceutical Sciences, Shenzhen University Health Science Center, Shenzhen 518060, China; yanym@szu.edu.cn (Y.-M.Y.); dweiwang@foxmail.com (D.-W.W.); 3Guangdong Key Laboratory for Functional Substances in Medicinal Edible Resources and Healthcare Products, School of Life Sciences and Food Engineer, Hanshan Normal University, Chaozhou 521041, China

**Keywords:** *Cyclopelta parva*, insects, xanthine analogues, guanosine derivatives

## Abstract

Five new compounds including three pairs of enantiomeric xanthine analogues, parvaxanthines D–F (**1**–**3**), two new guanosine derivatives, asponguanosines C and D (**6** and **7**), along with two known adenine derivatives were isolated from the insect *Cyclopelta parva*. Racemic **1**–**3** were further separated by chiral HPLC. Their absolute configurations were assigned by spectroscopic and computational methods. It is interesting that all of these isolates are natural product hybrids. Antiviral, immunosuppressive, antitumor and anti-inflammatory properties of all the isolates were evaluated.

## 1. Introduction

Natural products are important sources of bio-active drugs and play a significant role in the drug discovery and development process due to their structurally diverse and biological activities [1,2]. In the past decades, more attention has been given to microorganisms, and medicinal plants. However, chemical investigations on medicinal insects remain largely unknown [3,4]. Since ancient time, medicinal insects have been used to prevent and treat diseases in China. Whereas previous studies on insects mainly focused on biologically active proteins and peptides, little is known about their non-peptide natural products. In recent years, we have focused on unlocking small non-peptide molecules from medicinal insects. As a result, bio-active and structurally diverse substances have been characterized by us from several insects [5,6,7,8,9,10,11]. *Cyclopelta parva* is mainly distributed in southern China, such as in Hunan, Hubei, Guangdong and Yunnan provinces. Previous pharmacological studies revealed that *C. parva* possesses significant anticancer, antibacterial, antifatigue activities [12,13,14]. We have initiated an investigation on *Aspongopus chinensis* and identified structurally interesting substances [15,16,17], which belong to the same family as that of *C. parva*. These findings prompted us to conduct a study in *C. parva*. As a result, COX-2 inhibitory dihydroxydopamine derivatives have been isolated by our group [18]. As a part of our relay study on the title insect, five new (**1**–**3**, **6** and **7**) and two known (**4** and **5**) base or nucleoside conjugates were isolated. This paper describes their isolation, structural identification, and biological evaluation (Figure 1).

## 2. Results and Discussion

### 2.1. Structure Elucidation of the Compounds

Parvaxanthine D (**1**) was assigned as C_11_H_14_N_4_O_4_ by analysis of its HRESIMS (*m/z* 267.1078 [M + H]^+^ (calcd for C_11_H_15_N_4_O_4_, 267.1090), indicating 7 degrees of unsaturation. The ^13^C-NMR along with the DEPT 135 experimental data of **1** (Table 1) show 11 carbon signals ascribed to one methyl, three methylenes, two methines (one olefinic), and five quaternary carbons (three carbonyl, two sp^2^). The ^1^H-^1^H COSY correlations of H-2ʹ/H-3ʹ/H-4ʹ/H-5ʹ/H-6ʹ (Figure 2) alongside the HMBC correlation from H-2ʹ to C-1ʹ (Δ_C_ 173.4) (Figure 2) indicates the presence of a hexanoic acid residue. Apart from this hexanoic acid residue fragment, the remaining NMR signals resemble those in a xanthine [19]. Careful comparison of the NMR data of **1** with parvaxanthine A [18], it revealed that they have similar structures, the only difference is that the hexanoic acid residue is connected to xanthine via *N*-9, the HMBC correlations of H-3ʹ/C-4, C-8 supported this conclusion. This substance is a enantiomeric mixture, which was further purified by chiral HPLC to afford (+)-**1** (**1a**) and (−)-**1** (**1b**), and the absolute configurations were assigned as 3′*R* for **1a** by comparing the ECD spectrum of (3′*R*)-**1**, agreeing well with the experimental one of (+)-**1** (**1a**) (Figure 3). Thus, the structure of **1** was identified and named parvaxanthine D.

Parvaxanthine E (**2**), with the molecular formula C_12_H_16_N_4_O_4_, was established on the basis of its HRESIMS (*m/z* 281.1242 [M + H]^+^ (calcd for C_12_H_17_N_4_O_4_, 281.1240). Detailed analysis of the NMR data of **2** with those of **1** demonstrated that they have similar structures, except the presence of an additional methoxy group in **2**. The HMBC correlations of OCH_3_/C-1ʹ indicates the position of the CH_3_O group in **2**. Thus the planar structure of **2** was assigned. Compound **2** was isolated as a racemic mixture. This was confirmed also by chiral HPLC analysis, which afforded to (+)-**2** (**2a**) and (−)-**2** (**2b**) by a Chiralpak IC column. Their absolute configurations were assigned as 3ʹ*S* for **2b** by comparing its CD spectrum with the experimental one of (−)-**1** (**1b**) (Figure 3). In this way, the structure of **2** was identified and named parvaxanthine E.

Parvaxanthine F (**3**) was obtained as a white solid, and the molecule formula was determined as C_11_H_16_N_4_O_3_ (*m/z* 253.1293 [M + H]^+^ (calcd for C_11_H_17_N_4_O_3_, 253.1300) by using a combination of HRESIMS, ^13^C-NMR, and DEPT spectra. Compound **3** has a similar structure to parvaxanthine A [18], excepting that a caboxylic acid group in parvaxanthine A was replaced by a hydroxymethyl group in **3**, which gains supports from the ^1^H-^1^H COSY correlations of H-1ʹ (Δ_Ha_ 3.31; Δ_Hb_ 3.15)/H-2ʹ/H-3ʹ/H-4ʹ/H-5ʹ/H-6ʹ. **3** was also isolated as a racemic mixture, and chiral HPLC was performed to afford (+)-**3** (**3a**) and (−)-**3** (**3b**). The absolute configurations were defined as 3ʹ*S* for **3b** by comparing its CD spectrum with that of (−)-parvaxanthine A. The structure of **3** was therefore identified and named parvaxanthine F.

Asponguanosine C (**6**) has the molecular formula C_16_H_23_N_5_O_7_ deduced from the analysis of its HRESIMS (*m/z* 398.1674 [M + H]^+^ (calcd for C_16_H_24_N_5_O_7_, 398.1670), ^13^C-NMR, and DEPT spectra. The ^13^C-NMR and DEPT spectra give 16 carbon signals attributed to 1 × CH_3_, 4 × CH_2_ (three aliphatic), 6 × CH (one olefinic, five nitrogenated or oxygenated), and 5 × C (two carbonyl and three sp^2^). The ^1^H-^1^H COSY correlations of H-2″/H-3″/H-4″/H-5″/H-6″, in consideration of the chemical shift of C-1″ at Δ_C_ 175.2 and HMBCcorrelations of H-2″, H-3″/C-1″, indicate the presence of a hexanoic acid residue. In addition, the other NMR signals resemble a guanosine moiety, which is connected to C-3″ via a *N*H bridge, which is supported by the HMBC observation of H-3″/C-2. Therefore, the planar structure of **6** was established as shown. **6** was isolated as an optically pure compound evidenced by chiral HPLC analysis. For the sugar moiety of **6**, its configuration was determined to be d-ribose by acid hydrolysis followed by derivatization and comparison with the reference standard. With these data in hand, the absolute configuration of **6** at C-3″ was assigned as *S* by comparing the experimental ECD curve with the calculated one (Figure 4). As a result, the structure of **6** was deduced and named asponguanosine C.

Asponguanosine D (**7**) was found to have the molecular formula C_15_H_21_N_5_O_7_ deduced from its HRESIMS (*m/z* 406.1329 [M + Na]^+^ (calcd for C_15_H_21_N_5_O_7_Na, 406.1330), ^13^C-NMR, and DEPT spectra, indicating eight degrees of unsaturation. The ^1^H-NMR spectrum of **7** (Table 2) shows one proton signal at Δ_H_ 8.38 (s, H-8). The ^13^C-NMR and DEPT spectra reveal 15 carbon signals classified into 2 × CH_3_, 2 × CH_2_ (two oxygenated), 6 × CH (one olefinic, five nitrogen or oxygenated), and 5 × C (two carbonyls, three sp^2^ carbons). The NMR data of **7** are similar to those of **6**, except for a difference in the side chain. The ^1^H-^1^H COSY correlations of H-1″/H-2″ and H-4″/H-5″, and the HMBC correlations of H-1″, H-2″, and H-4″/C-3″, in consideration of the chemical shift of C-4″ (Δ_C_ 62.8), indicate the presence of an ethyl propionate side chain, which is connected to the guanosine residue via a *N*H bridge as suggested by the HMBC correlations of H-2″/C-2. **7** was also isolated as an optically pure compound. To clarify the absolute configuration of **7**, first hydrolysis followed by derivatization and comparison with the standards (d/l-ribose derivatives) allowed us to assign the sugar moiety in **7** as d-ribose. Next the ECD calculations were used to clarify the absolute configuration at C-2″. It was found that the ECD spectrum of (3″*S*)-**7** agrees well with the experimental one of **7** (Figure 4). As a consequence, the structure of **7** was identified as shown and named asponguanosine D.

It is interesting that all the isolates are natural product hybrids. Such a phenomenon has been found in *Aspongopus chinensis* [20]. In combination with the previous reports, it could be seen that alkylation of nitrogen is common in insect derived natural products. However, to the best of our knowledge, such compounds are not common in species beyond insects [21]. In addition, the reason that the insects synthesize these compounds by utilizing common nucleobases or nucleosides and a C6 unit remains unknown.

The known compounds (**4** and **5**) were identified to be aspongadenine A [15] and delicatuline B [22], respectively, by comparing their spectroscopic data with previously reported values.

### 2.2. Biological Evaluation

Natural product hybrids with profound biological activities have been described [23,24]. To explore the biological potential of such compounds, antiviral activity using vero cells, immunosuppressive activity in naive T cells, cytotoxic properties in human cancer cell lines (BGC-823, MDA-MB-231, HepG2, Kyse30), and LPS-induced pro-inflammatory expression in Raw264.7 cells were evaluated. Unfortunately, only compound **5a** was found to have anti-inflammatory activity, and the others are inactive.

To investigate whether isolated compounds have biological importance, the biological activities of the isolates were evaluated. All of the isolated compounds except **5a** were examined for their biologic activities on anti-inflammatory effects in RAW264.7 cells. To evaluate the cytotoxicity of compound **5a**, cells were incubated with different concentrations of compound **5a** (0 μM, 5 μM, 10 μM and 20 μM) for 24 h. It was found that compound **5a** did not affect the viability of the RAW264.7 cells by CCK-8 assay (Figure 5A). Then we measured the production of inflammatory cytokines treated with LPS in the presence or absence of compound **5a**. The ELISA assay showed that compound **5a** could suppress the production of IL-6 in LPS-stimulated RAW264.7 cells in a concentration-dependent manner (Figure 5B). In addition, since nucleobases, nucleosides, and the C6 unit are all common and structurally simple compounds, whether their hybrids play an essential role in ecological aspects needs further exploration.

## 3. Experimental Section

### 3.1. General Procedures

Optical rotations were recorded on a Horiba SEPA-300 polarimeter. NMR spectra were recorded on a Bruker AV-500 and AV-600 spectrometer (Bruker, Karlsruhe, Germany) with TMS as an internal standard. UV and CD spectra were obtained using a Chirascan instrument (Agilent Technologies, Santa Clara, CA, USA). HRESIMS were measured with a Shimadzu LC-20AD AB SCIEX triple TOF X500R MS spectrometer (Shimadzu Corporation, Tokyo, Japan). Macroporous adsorbents (Rohmhaas AMBERLITETM XAD 16N, America), Sephadex LH-20 (Amersham Biosciences, Uppsala, Sweden), YMC gel ODS-A-HG (40–60 μm; YMC Co., Japan), RP-18 silica gel (40–60 μm; Daiso Co., Japan) and MCI gel CHP 20P (75–150 μm, Mitsubishi Chemical Industries, Tokyo, Japan) were used for column chromatography. Preparative HPLC was taken on a Chuangxin-Tongheng Chromatograph equipped with a Thermo Hypersil GOLD-C18 column (250 mm × 21.2 mm, i.d., 5 μm). Semi-preparative HPLC separation were conducted on a Saipuruisi Chromatograph with a UV detector and a YMC-Pack ODS-A column (250 mm × 10 mm, i.d., 5 μm). Chiral separation was carried out on a chiral HPLC equipped with a UV detector and a Daicel Chiralpak Phenomenex column (OOG-4762-E0 LUX® i-Amylose-1, 250 mm × 4.6 mm, i.d., 5 μm) or a Daicel Chiralpak column (IC, 250 mm × 4.6 mm, i.d., 5.5 μm; OD-H, 250 mm × 4.6 mm, i.d., 5 μm; AD-H, 250 mm × 4.6 mm, i.d., 5 μm) at a flow rate of 1.0 mL/min.

### 3.2. Insect Material

*C. parva* was purchased and identified by Prof. De-Po Yang at School of Pharmaceutical Sciences, Sun Yat-Sen University. The material was purchased from Hunan Zhenxing Co. Ltd. for Chinese Materia Medica, Changsha, China, in December 2017. A voucher specimen (No. CHYX0628) is deposited at the School of Pharmaceutical Sciences, Shenzhen University, China.

### 3.3. Extraction and Isolation

The air-dried powdered *C. parva* (45 kg) was extracted with 80% aqueous EtOH (3 × 270 L, 24 h each) at room temperature. The combined extracts were concentrated to obtain a crude extract, which was suspended in water and successively partitioned with petroleum ether and ethyl acetate. The water-soluble extracts (4.0 kg) were divided into 6 parts (Fr.1–Fr.6) by using a macroporous adsorbents AMBERLITE XAD 16N column eluted with gradient aqueous EtOH (10:90–100:0). Fr.2 (170 g) was isolated by a MCI gel CHP 20P column eluted with gradient aqueous MeOH (2:98–60:40) to afford nine fractions (Fr.2.1–Fr.2.9). Fr.2.4 (25.8 g) was subjected to an ODS column on MPLC system eluted with gradient aqueous MeOH (2:98–60:40, flow rate: 20 mL/min) to afford seven fractions (Fr.2.4.1–Fr.2.4.7). Fr.2.4.3 (4.89 g) was subjected to gel filtration on Sephadex LH-20 (MeOH/H_2_O, 75:25) to give seven portions (Fr.2.4.3.1–Fr.2.4.3.7). Fr.2.4.3.5 (1.09 g) was separated by using a MCI gel CHP 20P column eluted with gradient aqueous MeOH (5:95–45:55) to provide nine portions (Fr.2.4.3.5.1–Fr.2.4.3.5.9). Fr.2.4.3.5.4 (284 mg) was separated by preparative HPLC (MeOH /H_2_O with 0.04% TFA, 3–15%, flow rate: 10 mL/min) to yield four subfractions (Fr.2.4.3.5.4.1–Fr.2.4.3.5.4.4). Fr.2.4.3.5.4.3 (95.6 mg) was purified by semi-preparative HPLC (MeOH/H_2_O with 0.04% TFA, 22:78, flow rate: 3 mL/min) to obtain **2** (t_R_ = 25.2 min, 4.30 mg) and **1** (t_R_ = 34.1 min, 12.31 mg). Compound **1** is a racemate that was separated by semi-preparative HPLC on a chiral phase by equipping with a Daicel Chiralpak IC (n-hexane/EtOH containing 0.04% TFA, 45:55, flow rate: 1.0 mL/min) to afford (+)-**1** (t_R_ = 9.4 min, 4.76 mg), **1a**; and (−)-**1** (t_R_ = 18.8 min, 4.87 mg), **1b**. Compound **2** is also a racemate, which was separated by semi-preparative HPLC on a chiral phase equipping with a Daicel Chiralpak IC (n-hexane/EtOH, 40:60, flow rate: 1.0 mL/min) to afford (+)-**2** (t_R_ = 15.0 min, 1.80 mg), **2a**; and (−)-**2** (t_R_ = 33.2 min, 1.84 mg), **2b**.

Fr.2.5 (26.8 mg) was divided into seven subfractions (Fr.2.5.1–Fr.2.5.7) by an ODS column on a MPLC system eluted with gradient aqueous MeOH (5:95–60:40, flow rate: 20 mL/min). Fr.2.5.4 (6.63 g) was separated by a RP-18 column (MeOH/H_2_O, 8:92–45:55) to give seven portions (Fr.2.5.4.1–Fr.2.5.4.7). Fr.2.5.4.3 (2.6 g) was subjected to gel filtration on Sephadex LH-20 (MeOH/H_2_O, 75:25) to give five portions (Fr.2.5.4.3.1–Fr.2.5.4.3.5). Fr.2.5.4.3.5 (167.3 mg) was purified by semi-preparative HPLC (MeOH/H_2_O, 28:72, flow rate: 3 mL/min) to obtain **3** (t_R_ = 27.1 min, 6.76 mg). Racemic **3** was separated by semi-preparative HPLC on a chiral phase by equipping with a Daicel Chiralpak OD-H (n-hexane/^i^PrOH with 0.04% TFA, 85:15, flow rate: 1.0 mL/min) to afford (+)-**3** (t_R_ = 17.5 min, 3.54 mg), **3a**; and (−)-**3** (t_R_ = 23.6 min, 2.71 mg), **3b**. Fr.2.5.4.4 (1.86 g) was separated by using Sephadex LH-20 (MeOH/H_2_O, 70:30) followed by semi-preparative HPLC (MeCN/H_2_O with 0.04% TFA, 11:89, flow rate: 3 mL/min) to obtain **6** (t_R_ = 20.6 min, 3.41 mg) and **7** (t_R_ = 24.8 min, 2.08 mg). Fr.2.5.3 (6.3 g) was separated by using Sephadex LH-20 (MeOH/H_2_O, 75:25) followed by a RP-18 column (MeOH/H_2_O, 5:95–35:65) to produce three portions (Fr.2.5.3.1–Fr.2.5.3.3). Fr.2.5.3.3 (2.0 g) was separated by using Sephadex LH-20 (MeOH/H_2_O, 70:30) followed by semi-preparative HPLC (MeOH/H_2_O with 0.04% TFA, 20:80, flow rate: 3 mL/min) to obtain **4** and **5**. Compound **4** is a racemate that was separated by semi-preparative HPLC on a chiral phase equipped with a Daicel Chiralpak IC (n-hexane/EtOH, 65:35, flow rate: 1.0 mL/min) to afford (+)-**4** (t_R_ = 15.3 min, 0.88 mg), **4a**; and (−)-**4** (t_R_ = 16.4 min, 1.3 mg), **4b**. Compound **5** is a racemate that was separated by semi-preparative HPLC on a chiral phase equipped with a Daicel Chiralpak IC (n-hexane/EtOH, 30:70, flow rate: 1.0 mL/min) to afford (+)-**5** (t_R_ = 6.3 min, 2.72 mg), **5a**; and (−)-**5** (t_R_ = 9.3 min, 1.98 mg), **5b** respectively.

### 3.4. Compound Characterization Data

Parvaxanthine D (**1**): white power; UV (MeOH) *λ*max (log*ε*) 261 (3.93), 235 (3.89), 200 (4.20) nm; {[α]^20^_D_ − 11.8 (*c* 0.03, MeOH); CD (MeOH) Δ*ε*_209_ + 1.15, Δ*ε*_233_ − 1.42; **1a**}; {[α]^20^_D_ + 6.3 (*c* 0.03, MeOH); CD (MeOH) Δ*ε*_209_ − 0.86, Δ*ε*_233_ + 1.03; **1b**}; HRESIMS (*m/z* 267.1078 [M + H]^+^ (calcd for C_11_H_15_N_4_O_4_, 267.1090); ^1^H and ^13^C-NMR data, see Table 1.

Parvaxanthine E (**2**): white solid; UV (MeOH) *λ*max (log*ε*) 258 (3.99), 241 (3.93), 200 (4.34) nm; {[α]^20^_D_ − 3.2 (*c* 0.03, MeOH); CD (MeOH) Δ*ε*_208_ + 1.10, Δ*ε*_233_ − 1.23; **2a**}; {[α]^20^_D_ + 6.7 (*c* 0.02, MeOH), CD (MeOH) Δ*ε*_208_ − 1.26, Δ*ε*_233_ + 0.45; **2b**}, HRESIMS (*m/z* 281.1240 [M + H]^+^ (calcd for C_12_H_17_N_4_O_4_, 281.1244); ^1^H and ^13^C-NMR data, see Table 1.

Parvaxanthine F (**3**): white solid; UV (MeOH) *λ*max (log*ε*) 269 (3.70), 201 (4.22) nm; {[α]^20^_D_ + 7.7 (*c* 0.03, MeOH); CD (MeOH) Δ*ε*_214_ + 0.42; **3a**}; {[α]_D_^20^ − 10.0 (*c* 0.02, MeOH); CD (MeOH) Δ*ε*_214_ − 0.20; **3b**}; HRESIMS (*m/z* 253.1296 [M + H]^+^ (calcd for C_11_H_17_N_4_O_3_, 253.1295); ^1^H and ^13^C-NMR data, see Table 1.

Asponguanosine C (**6**): yellow gum; UV (MeOH) *λ*max (log*ε*) 255 (3.80), 200 (4.16) nm; {[α]^20^_D_ + 108.3 (*c* 0.02, MeOH); CD (MeOH) Δ*ε*_201_ − 2.33, Δ*ε*_216_ + 1.48, Δ*ε*_249_ − 1.41, Δ*ε*_281_ + 0.58; **6**}; HRESIMS (*m/z* 398.1675 [M + H]^+^ (calcd for C_16_H_24_N_5_O_7_, 398.1670); ^1^H and ^13^C-NMR data, see Table 2.

Asponguanosine D (**7**): yellow gum; UV (MeOH) *λ*max (log*ε*) 254 (3.94), 200 (4.25) nm; {[α]^20^_D_ − 17.4 (*c* 0.02, MeOH); CD (MeOH) Δ*ε*_200_ − 1.90, Δ*ε*_214_ + 1.54, Δ*ε*_276_ − 0.38; **7**}; HRESIMS (*m/z* 406.1329 [M + Na]^+^ (calcd for C_15_H_21_N_5_O_7_Na, 406.1333); ^1^H and ^13^C-NMR data, see Table 2.

### 3.5. Acid Hydrolysis and Preparation of Sugar Derivatives of Compounds ***6*** and ***7***

Compounds **6** or **7** (each 0.5 mg) was submitted to hydrolysis with 0.8 mL of 6 N HCl at 60 °C for 1.5 h. The reaction mixtures were concentrated and l-cysteine methyl ester in pyridine (0.8 mL) solution was added to it and reflected at 60 °C for 1 h. Then 2-methylphenyl isothiocyanate was added to the reaction mixtures and further heated for one more hour. In the same way, the standards ribose d and l were also derivatized and the reaction mixtures were directly analyzed by RP-18 HPLC [YMC-Pack ODS-A column (250 mm × 4.6 mm, i.d., 5 μm); MeOH/H_2_O (0.05% CF_3_COOH), 40:60; flow rate: 0.8 mL/min]. The retention times (t*_R_*) of the derivatives of standard d-ribose and l-ribose were determined at 16.8 min and 13.3 min, respectively. By the comparison of retention times (t*_R_*) of compounds **6** and **7** derivatives with the standards (d/l-ribose derivatives), the ribose in compounds **6** and **7** were determined to be d-form (see Appendix A) [25,26].

### 3.6. Computational Methods

Molecular Merck force field (MMFF) and DFT/TDDFT were calculated with Spartan’14 software package and Gaussian 09 program package. Electronic circular dichroism (ECD) calculations were conducted at the B3LYP SCRF (PCM)/6-311G(d,p) level and the CD spectra were produced by the program SpecDis 1.62 [27].

### 3.7. Antiviral Activity Assay

Antiviral activity [28] was determined by plaque assay using monolayer cultures of African green monkey kidney (Vero) cells in 96-well culture plates with 1 × 10^4^ cells/well. After 24-h incubation, the growth medium (RPMI-1640 medium supplemented with 10% Fetal Bovine Serum (FBS)) was discarded, and the cells incubated with compounds were diluted to 3 series concentrations (50.00, 25.00, and 12.50 μM) in maintenance solution (RPMI-1640 medium supplemented with 2% FBS). The experiment was repeated three times with 7 replicates for each concentration. The negative control wells represented Vero cells with maintenance medium without virus. All cells were cultured at 37 °C in a 5% CO_2_ atmosphere for 72 h and the cytopathic condition was observed under an inverted microscope. When the cytopathic effect in the virus control group (positive control) reached more than 75%, we observed and recorded the cytopathic effect (CPE) of each well (see Appendix A).

### 3.8. Immunosuppressive Activity Assay

The cytotoxicity of the compounds (20 μM) on resting lymphocytes from C57/BL6 mice was determined. Spleen cells (5 × 10^5^/well) were incubated for 24 h in the presence of various concentrations of compounds. Cell viability was measured at 570 nm by MTT uptake. T lymphocytes obtained from C57/BL6 mice were cultured with compounds (20 μM) for 4 h, then treated with murine IFN-γ (5 ng/mL) for 30 min. After the incubation, proteins were extracted and assessed by Western blot analysis. Mouse T lymphocytes (2 × 10^5^/well) obtained from splenocytes of C57/BL6 mice were stimulated by Con A (5 μg/mL) for 48 h, in the presence of compounds (20 μM) and their proliferation was evaluated using MTT assay (see Appendix A) [29,30].

### 3.9. Biological Evaluation for Human Cancer Cells (BGC-823, MDA-MB-231, HepG2, Kyse30)

All the cell lines were purchased from the Cell Bank of China Science Academy and the cell lines were incubated at 37 °C under 5% CO_2_ atmosphere by using Dulbecco’s modified Eagle’s medium (DMEM) supplemented with 100 U/ml penicillin-streptomycin and 10% FBS. Cell viability was evaluated with Cell Count Kit-8 (CCK-8) assay kit and exponentially growing cells were seeded at 3–8 × 10^3^ cells per well in 96-well culture plates for 24 h. Cells were treated with increasing concentrations (up to 40 μM) of these compounds for 48 h and an equal volume of DMSO was used as a control. CCK-8 solution (10 μL) was added to each well and the light absorbance was measured at 450 nm after incubation for another 0.5–4 h. The cell survival values were determined by the PrismPad program [31,32]. Cytotoxic effects of compounds were determined at 40 μM for 48 h in human cancer cells (BGC-823, MDA-MB-231, HepG2, Kyse30) and the effects of all the compounds were determined by CCK-8 assay (see Appendix A).

### 3.10. Anti-Inflammatory Activity

RAW264.7, a mouse macrophage cell line, was cultured in high-glucose DMEM supplemented with 10% FBS, 100 μg/mL streptomycin and 100 U/mL penicillin at 37 °C in a humidified environment containing 5% CO_2_. RAW264.7 (2 × 10^4^ cells/mL) cells were cultivated overnight into 96-well plates with completed DMEM and treated with DMSO or various concentrations of compounds for 24 h. Then, 10 μL CCK-8 solution was added into each well for 1 h at 37 °C and the absorbance of each well was recorded at 450 nm using a microplate reader. The cells (RAW264.7) were pre-treated with different concentrations of compounds for 2 h and then stimulated with LPS (1 μg/mL) for an additional 4 h. The culture supernatant from each well was collected and assayed with IL-6 ELISA Kit.

## 4. Conclusions

Herein we report the isolation and structural elucidation of five new and two known metabolites from the insect *C. parva*. In addition, antiviral, immunosuppressive, anti-tumor and anti-inflammatory properties of all the isolates were determined. We concluded that only compound **5a** could suppress the production of IL-6 in LPS-stimulated RAW264.7 cells in a concentration dependent manner.

## Figures and Tables

**Figure 1 molecules-26-03531-f001:**
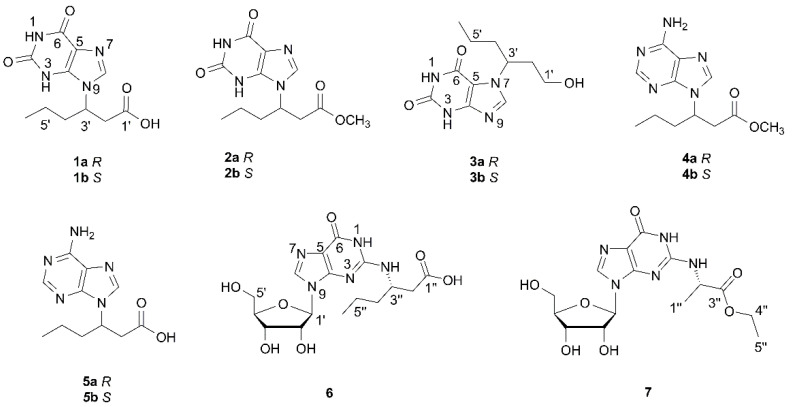
The structures of compounds **1**–**7** from *Cyclopelta parva*.

**Figure 2 molecules-26-03531-f002:**
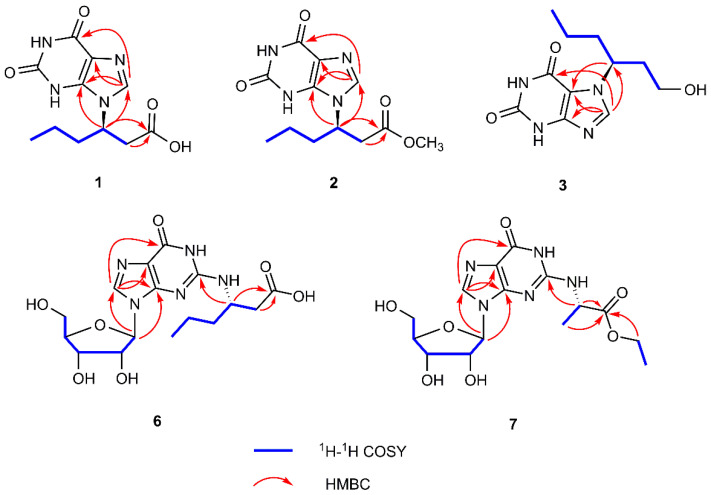
Key ^1^H-^1^H COSY and HMBC correlations for **1**–**3** and **6**, **7**.

**Figure 3 molecules-26-03531-f003:**
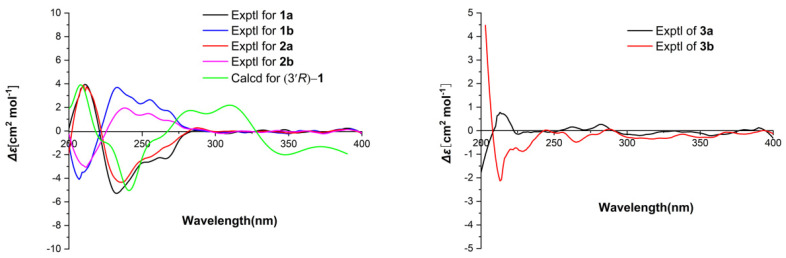
Comparison of the calculated ECD spectrum for (3ʹ*R*)-**1** at B3LYP/6-311g(d,p) level with the experimental spectra of **1a**, 1**b**, **2a**, 2**b** in MeOH (**left**), σ = 0.2 eV, shift = −10 nm. The experimental spectra of **3a**,3**b** in MeOH (**right**).

**Figure 4 molecules-26-03531-f004:**
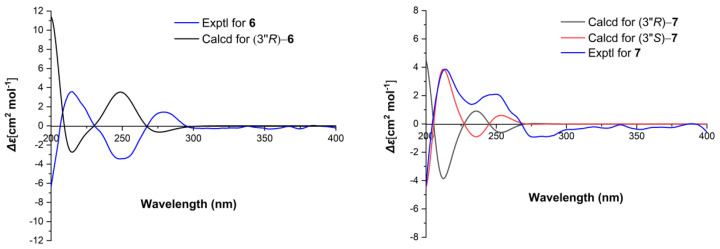
Comparison of the calculated ECD spectrum for (3″*R*)–**6** with the experimental spectrum of **6** at B3LYP/6-31(d,p) level in MeOH (**left**). σ = 0.3 eV, shift = +5 nm. Comparison of the calculated ECD spectra for (3″*R*)–**7** and (3″*S*)–**7** with the experimental spectrum of **7** at B3LYP/6-31(d,p) level in MeOH (**right**). σ = 0.16 eV, shift = 0 nm.

**Figure 5 molecules-26-03531-f005:**
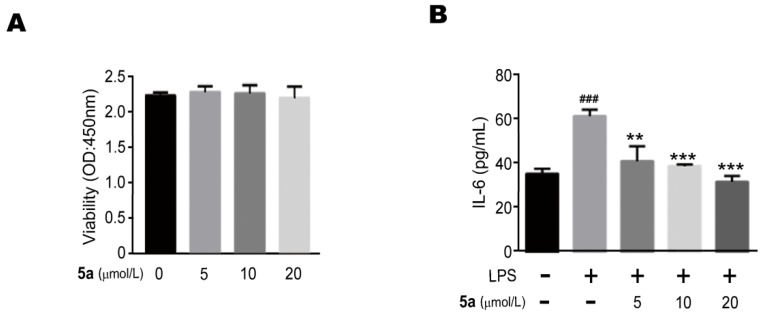
Compound **5a** increased the effects of the pro-inflammatory cytokine IL-6. RAW264.7 cell proliferation in response to compound **5a** at different doses assayed by CCK-8 assay (**A**). Compound **5a** suppressed LPS-induced IL-6 expression in RAW 264.7 cells. The cells were pretreated with different concentrations of compound **5a** for 2 h and then stimulated with 1 μg/mL LPS for 4 h. Culture media were collected in order to measure IL-6 concentrations using an ELISA kit. Data represent mean ± SEM values of three experiments. ** *p* < 0.01 and *** *p* < 0.001 compared with LPS alone. ### *p* < 0.001 compared with DMSO alone (**B**).

**Table 1 molecules-26-03531-t001:** ^1^H- (600 MHz) and ^13^C-NMR (150 MHz) data of **1**–**3** (Δ in ppm, *J* in Hz).

Position	1 ^a^		2 ^b^		3 ^b^	
	Δ_H_	Δ_C_	Δ_H_	Δ_C_	Δ_H_	Δ_C_
2		152.8, s		150.9, s		150.2, s
4		142.6, s		140.6, s		151.1, s
5		115.9, s		114.7, s		106.2, s
6		159.9, s		157.9, s		155.4, s
8	7.99 (s)	136.4, d	7.89 (s)	134.6, d	8.03 (s)	142.5, d
1′a		173.4, s		170.5, s	3.31 (m)	57.3, t
1′b					3.15 (m)	
2′a	2.99 (dd, 17.0, 9.3)	40.2, t	3.05 (dd, 17.0, 9.4)	38.4, t	2.12 (m)	37.2, t
2′b	2.97 (dd, 17.0, 5.4)		3.00 (dd, 17.0, 5.2)		1.98 (m)	
3′	4.70 (m)	54.5, d	4.71 (m)	51.6, d	4.63 (m)	55.1, d
4′	1.91 (m)	38.0, t	1.76 (m)	36.3, t	Ha: 1.90 (m)Hb: 1.70 (m)	36.2, t
5′a	1.29 (m)	20.2, t	1.18 (m)	18.4, t	1.10 (m)	18.7, t
5′b	1.18 (m)		1.00 (m)		1.02 (m)	
6′	0.93 (t-like, 7.3)	13.9, q	0.81 (t-like, 7.3)	13.4, q	0.82 (t-like, 7.3)	13.5, q
OCH_3_			3.51 (s)	51.5, q		

^a^ In methanol-*d*_4_, ^b^ In DMSO-*d*_6._

**Table 2 molecules-26-03531-t002:** ^1^H- (600 MHz) and ^13^C-NMR (150 MHz) data of **6** and **7** in methanol-*d*_4_ (Δ in ppm, *J* in Hz).

Position	6		7	
	Δ_H_	Δ_C_	Δ_H_	Δ_C_
2		154.2, s		153.7, s
4		152.2, s		152.1, s
5		115.9, s		117.2, s
6		158.3, s		158.7, s
8	8.45 (s)	138.1, d	8.38 (s)	138.2, d
1ʹ	5.93 (brd, 4.4)	90.5, d	5.91 (brd, 4.6)	89.8, d
2ʹ	4.64 (t-like, 5.1)	75.6, d	4.54 (t-like,5.0)	75.9, d
3ʹ	4.33 (t-like, 5.1)	71.6, d	4.30 (t-like, 5.0)	71.8, d
4ʹ	4.10 (m)	86.7, d	4.08 (m)	86.7, d
5ʹ	Ha: 3.88 (dd, 12.1, 3.2); Hb: 3.77 (dd, 12.1, 3.9)	62.7, t	Ha: 3.85 (dd, 12.1, 3.1); Hb: 3.74 (dd, 12.1, 3.7)	62.8, t
1″		175.2, s	1.51 (d, 7.2)	18.1, q
2″	Ha: 2.62 (dd, 15.8, 5.8); Hb: 2.59 (dd, 15.8, 6.1)	39.8, t	4.62 (q, 7.2)	51.2, d
3″	4.42 (m)	49.0, d		174.8, s
4″	1.65 (m)	37.6, t	4.23 (m)	62.8, t
5″	1.45 (m)	20.4, t	1.29 (t, 1.7)	14.6, q
6″	0.97 (t-like, 7.4)	14.3, q		

## Data Availability

Data is contained within the article or supplementary material.

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
