# Peer review of "Parvaxanthines D–F and Asponguanosines C and D, Racemic Natural Hybrids from the Insect Cyclopelta parva"

_molecules, 2021, doi:10.3390/molecules26123531_

Round 1
Reviewer 1 Report
Authors report the structures of parvaxanthines and asponguanosines. The structural elucidation is well explained. Therefore, these results are worthy being published in Molecules.
- The compounds 4 and 5 are included in Figure 1, but authors did not explain those were known compounds. It is very confusing. Authors must describe the compounds 4 and 5 are known in Introduction or figure caption of Figure 1.
- Page 2, Line 75. H-1’ (1.90; 1.70) /H-2’/..... The chemical shifts of H1’ are 3.31 and 3.15.
- The absolute configuration of the sugar moiety in compound 7 is not shown in Figure 1.
- Page 2, Line 48. “hexanoc acid”
- Page 5, Line 145. “activities of were “
Author Response
Response to Reviewer 1 Comments:
Thank you very much for your valuable suggestions
Point 1: The compounds 4 and 5 are included in Figure 1, but authors did not explain those were known compounds. It is very confusing. Authors must describe the compounds 4 and 5 are known in Introduction or figure caption of Figure 1.
Response 1: Thank you for your advice. I have added descriptions of compounds 4 and 5 are known in introduction, and in line 132. I have modified as “The known compounds (4 and 5) were identified to be aspongadenine A[15], del-icatuline B[22] respectively by comparing their spectroscopic data with previously reported values”.
Point 2: Page 2, Line 75. H-1ʹ (1.90; 1.70) /H-2ʹ/..... The chemical shifts of H1ʹ are 3.31 and 3.15.
Response 2: Thank you for your advice. I have corrected it.
Point 3: The absolute configuration of the sugar moiety in compound 7 is not shown in Figure 1.
Response 3: Thank you for your advice. I have corrected it and showed in figure 1.
Point 4: Page 2, Line 48. “hexanoc acid”.Page 5, Line 145. “activities of were”
Response 4: We thank that you read this paper with great patient, we have checked and rewrote it, and modified as“hexanoic acid”. “therefore the biological activities of isolates were evaluated”.
Reviewer 2 Report
Dear authors: the manuscript describes a remarkable amount of work and there are minor suggestions to improve the presentation of the manuscript. Attached you will find the main body of the manuscript in PDF with comments added as well as the supplementary material section with minor suggestions added.

Author Response
Response to Reviewer 2 Comments
Thank you very much for your valuable suggestions
Point 1: Figure S3: “add chemical structure within available space in HSQC spectra”. Figure S10: “a exiplicit assignment is recommended to facilit ate inspection by readers”.
Response 1: Thank you for your advice. We have added chemical structure within available space in HSQC spectra in Figure S3, and we have marked the chemical shifts of all signals in Figure S10.
Point 2: Increase color or contrast to improve visibility.
Response 2: Thank you, Figure S43 was provided again. Because the acid hydrolysis experiment was done last year, due to the small amount of compounds, we cannot repeat the experiment and provide new analysis graphs. So it can only increase color or contrast to improve visibility. I have set to display two wavelength absorption (210 nm (red trace) and 254 nm (black trace)).
Point 3: Figure 1: “include representation of stereochemistry as compound 6 dose”.
Response 3: Thank you for your advice, but I think this way of presentation is no problem. Because compound 1−3 are racemic mixtures, we obtained the enantiomers after using chair HPLC, and they have same planar structure. Previously published articles (Di, L.; Shi, Y.N.; Yan, Y.M.; et al. Nonpeptide small molecules from the insect Aspongopus chinensis and their neural stem cell proliferation stimulating properties. RSC Adv. 2015, 5, 70985-70991, doi: 10.1039/c5ra12920f.) also used this way, and several reviewers also suggested that we should use the current expression because it is more concise. If I use other methods of expression, it will be cumbersome.
Point 4: Figure 2: include the graphical representation of stereochemistry at C3.
Response 4: In Figure 2, the purpose is to express the key 1H-1H COSY and HMBC correlations for 1–3 and 6–7, and compound 1−3 are racemic mixtures (a/b), I think it is not necessary to include a graphical representation of the stereochemistry at C3. Our previous published articles also used this way.
Point 5: Explain the meaning of “planar structure”.
Response 5: “Planar structure of compound was assigned” means by using what ever we used techniques (HRMS and NMR) for characterization, the structure (planar) was assigned with out relative or absolute configurations. After getting planar structure we use further techniques for assigning of absolute stereo chemistry.
Point 6: “whichwas”.“were undergo hydrolysis”.“concluded only”.
Response 6: “whichwas” modified as “which was” are” as suggested by reviewer #2.
“were undergo hydrolysis” modified as “were submittded to hydrolysis” are” as suggested by reviewer #2.
“concluded only” modified as “concluded that only” are” as suggested by reviewer #2.
“following are” modified as “following materials are” as suggested by potential reviewer #2.
This manuscript is a resubmission of an earlier submission. The following is a list of the peer review reports and author responses from that submission.